# Transport and magnetic properties in the Nd diluted system $Y_{1-x}Nd_xCo_2Zn_{20}$

**Rikako Yamamoto[1⋆], Yasuyuki Shimura[1], Kazunori Umeo[2], Toshiro Takabatake[1] and Takahiro Onimaru[1]**

**1** Department of Quantum Matter, Graduate School of Advanced Science and Engineering, Hiroshima University, Higashi-Hiroshima 739-8530, Japan
**2** Department of Low-Temperature Experiment, Integrated Experimental Support/Research Division, N-BARD, Hiroshima University, Higashi-Hiroshima 739-8526, Japan

⋆ ryama@hiroshima-u.ac.jp

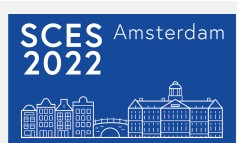 *International Conference on Strongly Correlated Electron Systems (SCES 2022) Amsterdam, 24-29 July 2022* doi:10.21468/SciPostPhysProc.11

## Abstract

We report the electrical resistivity, specific heat, and magnetization measurements of $Y_{1-x}Nd_xCo_2Zn_{20}$ for $0.017 \leq x \leq 0.95$. The Schottky-type specific heat peak at around 13 K for all the samples is reproduced by the crystalline electric field model with the $\Gamma_6$ doublet ground state of a $Nd^{3+}$ ion. The magnetization and magnetic susceptibility data of the samples for $x \leq 0.06$ are well reproduced by the calculation without intersite magnetic interactions among Nd moments. Therefore, the dilute Nd system $Y_{1-x}Nd_xCo_2Zn_{20}$ for $x \leq 0.06$ is a good candidate to study on-site interaction of the $\Gamma_6$ doublet ground state of $4f$ electrons with conduction electrons.



## 1 Introduction

The caged compounds $RTr_2X_{20}$ ($R$: rare-earth, $Tr$: transition metal, $X$ = Al, Zn, and Cd) crystallize in the cubic CeCr₂Al₂₀-type structure with the space group of $Fd\bar{3}m$ (No. 227, $O_h^7$) [1]. The $R^{3+}$ ions at the $8a$ site with the cubic point group $T_d$ are encapsulated in the Frank-Kasper cages formed by sixteen $X$ atoms. This feature weakens the crystalline electric field (CEF) effect and enhances hybridization of $4f$ electrons with conduction electrons ($c-f$ hybridization).

In a Pr-based compound PrIr₂Zn₂₀, this characteristic gives rise to non-Fermi liquid (NFL) behavior related to the quadrupolar degrees of freedom in the $\Gamma_3$ doublet ground state of a $Pr^{3+}$ ion under the cubic CEF [2]. In fact, the temperature dependences of the electrical resistivity $\rho$ and the magnetic specific heat $C_m$ agree with those calculated with the two-channel Anderson

lattice model. Therefore, formation of a quadrupolar Kondo lattice was proposed [3]. In recent years, a Pr-diluted system $Y_{1-x}Pr_xIr_2Zn_{20}$ has been systematically studied to investigate the single-site quadrupolar Kondo effect [4–6]. In addition to the NFL behaviors of $\rho(T)$ and $C_m(T)$, the elastic constant $(C_{11}-C_{12})/2$ shows a logarithmic temperature dependence below 0.3 K, providing another support for the single-site quadrupolar Kondo effect. On the other hand, the quadrupolar Kondo effect predicted the residual entropy of $0.5R\ln2$ at $T = 0$, which has not been observed yet.

The Nd-based family $NdTr_2Zn_{20}$ ($Tr = $ Co, Ru, Rh, Os, and Ir) and $NdTr_2Al_{20}$ ($Tr = $ Ti, V, and Cr) with mostly the $\Gamma_6$ doublet ground state of the $4f^3$ configuration provide a new platform to investigate the magnetic two-channel Kondo effect. A theoretical calculation using a numerical renormalization group method with a seven-orbital impurity Anderson model showed that the residual entropy of $0.5R\ln2$, which is the characteristic of the two-channel Kondo effect, manifests itself in a wide range of parameters for the local $\Gamma_6$ doublet ground state [7]. Here, it is noted that relatively large $c-f$ hybridization is needed to exhibit the two-channel Kondo effect in the $4f^3$ systems. Among $NdTr_2X_{20}$, the $c-f$ hybridization in $NdCo_2Zn_{20}$ is expected to be larger than the other $NdTr_2X_{20}$ compounds, because the lattice parameter is the smallest and the magnetic transition temperature $T_N = 0.53$ K is the lowest [8–13]. In fact, the $\rho(T)$ data of $NdCo_2Zn_{20}$ decrease with an upward convex curvature on cooling from 4 K to $T_N$, which is expressed by the theoretical form derived from the two-channel Anderson lattice model [8]. However, it is not clear whether this temperature variation of $\rho(T)$ is ascribed to the two-channel Kondo effect or intersite magnetic interaction between Nd moments.

In this paper, we focus on $Y_{1-x}Nd_xCo_2Zn_{20}$ for $0.017 \leq x \leq 0.95$ to study how the intersite magnetic interaction is modified by the Nd substitutions. In intermetallic compounds, the Ruderman-Kittel-Kasuya-Yosida (RKKY) interaction is mediated by the spin polarization of conduction electrons, which sign oscillates with respect to the distance. The oscillating and long-ranged nature is well manifested in $La_{1-x}Nd_x$ [14]. An antiferromagnetic order in $x = 1$ changes to a ferromagnetic one for $x = 0.6$ and it persists down the to $x = 0.2$. These results indicate that the intersite magnetic interaction in the Nd rich region depends on band structure. In analogy, to examine the on-site interaction of the $4f^3$ state with conduction electrons in $Y_{1-x}Nd_xCo_2Zn_{20}$, the dilute range $x < 0.2$ should be studied carefully. We synthesized the single-crystalline samples of $Y_{1-x}Nd_xCo_2Zn_{20}$ and measured the electrical resistivity, specific heat, isothermal magnetization, and magnetic susceptibility for $T \geq 1.8$ K.

## 2 Experimental procedure

Single-crystalline samples of $Y_{1-x}Nd_xCo_2Zn_{20}$ for $0.017 \leq x \leq 0.95$ were synthesized by the Zn self-flux method as described in the previous report [15]. We used high purity elements of Y (99.9%), Nd (99.99%), Co (99.9%), and Zn (99.9999%). The samples were characterized by powder x-ray diffraction (XRD) measurements. The Rietveld analysis of the XRD patterns using RIETAN-FP [16] confirmed the $CeCr_2Al_{20}$-type structure for all the samples. The lattice parameter determined by the Rietveld analysis increases linearly with respect to $x$. The atomic compositions were obtained by the wavelength-dispersive electron-probe microanalysis (EPMA). For the Nd diluted samples with $x < 0.05$, it was difficult to determine the compositions of the bulk samples since the EPMA probes only the surface and the resolution is not high for determining the small ratio below 0.05. To estimate the Nd compositions more accurately, we measured the magnetization curves at $T = 1.8$ K to compare with that calculated by using the CEF level schemes determined by inelastic neutron scattering (INS) measurements of $NdCo_2Zn_{20}$ [17].

The electrical resistance was measured by a standard four-probe AC method with a laboratory-built system using a Gifford-McMahon-type refrigerator in the temperature range of $3-300$ K. Heat capacity measurements were performed by the thermal relaxation method between 4 and 300 K using a physical property measurement system (PPMS, Quantum Design). Magnetization measurements were carried out with a commercial superconducting quantum interference device magnetometer (MPMS, Quantum Design) from 1.8 to 300 K in magnetic fields up to 5 T.

## 3 Results and discussion

Figure 1(a) shows the temperature dependences of the normalized electrical resistivity $\rho(T)/\rho(300\,\text{K})$ of $Y_{1-x}Nd_xCo_2Zn_{20}$ including end compositions $x = 0$ and 1 [8]. The electric current was applied along the [100] direction for the single-crystalline samples. The data decrease with downward curvature, and asymptotically approach constant values below 10 K. As shown in the inset of Fig. 1, the residual resistivity ratio defined as RRR $= \rho(300\,\text{K})/\rho(3\,\text{K})$ largely decreases from 81.4 for $x = 0$ to 13.4 for $x = 0.017$.

The specific heat $C$ versus temperature is displayed in Fig. 1(b). The $C(T)$ data certainly reach the Dulong-Petit value of 573.7 J/K mol at 300 K as expected for a compound with 23 atoms in the formula unit. The magnetic specific heat divided by temperature as a function of temperature, $C_m(T)/T$ vs $T$, is shown in the inset of Fig. 1(b). We obtained the magnetic contribution $C_m(T)$ by subtracting the $C(T)$ data of a nonmagnetic counterpart $YCo_2Zn_{20}$ as the lattice contribution from the measured specific heat. The $C_m(T)/T$ data show maxima at around 13 K, representing the Schottky specific heat due to the thermal excitations from the CEF ground state to the excited CEF levels. The CEF Hamiltonian $\mathcal{H}_{\text{CEF}}$ for the $Nd^{3+}$ ion under the cubic CEF is described as [18]

$$\mathcal{H}_{\text{CEF}} = W\left[\frac{X}{60}(O_4^0 + 5O_4^4) + \frac{1-|X|}{2520}(O_6^0 - 21O_6^4)\right]. \tag{1}$$

The solid line is the CEF calculation by using the CEF level scheme of $\Gamma_6(0\text{ K})-\Gamma_8^{(1)}(44\text{ K})-$

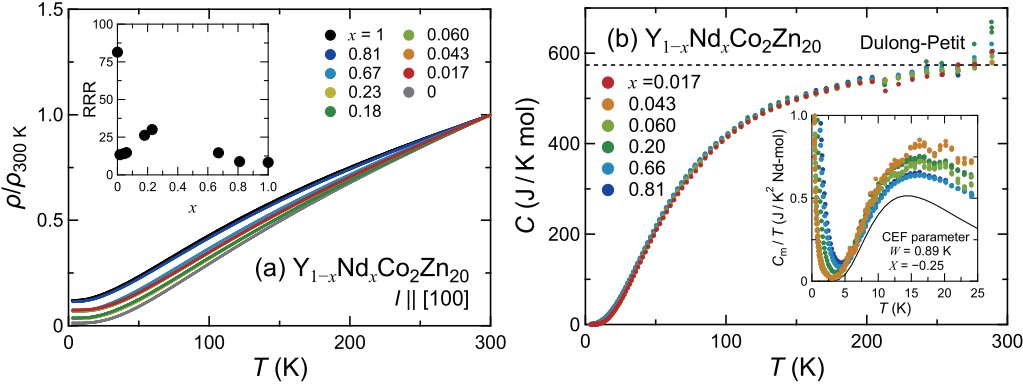

Figure 1: (a) Normalized electrical resistivity $\rho(T)/\rho(300\,\text{K})$ versus temperature $T$ of $Y_{1-x}Nd_xCo_2Zn_{20}$ for $0 \leq x \leq 1$. The $\rho(T)$ data for $x = 1$ are cited from [8]. The inset shows the residual resistivity ratio as RRR $= \rho(300\,\text{K})/\rho(3\,\text{K})$. (b) Specific heat $C$ versus $T$ of $Y_{1-x}Nd_xCo_2Zn_{20}$ for $0.017 \leq x \leq 0.81$. The dashed line is the Dulong-Petit value of 573.7 J/K mol$^{-1}$. The inset shows the magnetic specific heat divided by temperature $C_m/T$ for $x = 0.043$, 0.60, 0.20, 0.66, and 0.81. The solid line represents the CEF calculation.

$\Gamma_8^{(2)}$(84 K) for NdCo$_2$Zn$_{20}$ determined by the INS measurements [17]. Here, we adopted the CEF parameters of $W = 0.89$ K and $X = -0.25$. The maxima in $C_{\mathrm{m}}(T)/T$ stays at around 13 K as calculated by the CEF model. This fact indicates that the CEF level scheme hardly changes among Y$_{1-x}$Nd$_x$Co$_2$Zn$_{20}$.

Figure 2(a) shows the $4f$ contribution of the magnetic susceptibility $\chi_{\mathrm{Nd}}$ measured in $B = 0.1$ T applied along the [100] direction. The data for $x \leq 0.06$ was measured in $B = 0.5$ T. The $\chi_{\mathrm{Nd}}$ data are deduced by subtracting the $\chi(T)$ data of YCo$_2$Zn$_{20}$ from the measured magnetic susceptibility. It is noted that the band structure calculation of YCo$_2$Zn$_{20}$ suggests an intermetallic state with negligible electron-electron correlation, leading to a non-magnetic ground state [19, 20] The $\chi_{\mathrm{Nd}}$ data follow the Curie–Weiss law above 50 K. The effective magnetic moments are estimated to be 3.7–4.2 $\mu_{\mathrm{B}}$/Nd, which are slightly higher than 3.62 $\mu_{\mathrm{B}}$ for a free Nd$^{3+}$ ion. The paramagnetic Curie temperatures $\theta_{\mathrm{p}}$ are negative, indicating that the antiferro-type magnetic interaction is predominant. Below 10 K, the $\chi_{\mathrm{Nd}}$ data depends on $x$. The $\chi_{\mathrm{Nd}}$ data for $x \geq 0.19$ are larger than the solid line that was calculated by the CEF model without a molecular-field parameter. On the other hand, as $x$ is reduced from $x = 0.19$, the $\chi_{\mathrm{Nd}}$ data for $T < 10$ K approach the solid line. Considering that the $\Gamma_6$ doublet ground states is mostly populated below 10 K, the intersite magnetic interaction among the CEF ground states is ferro-type but negligible for $x \lesssim 0.06$.

The strength of intersite magnetic interaction was estimated from mean-field analysis of the isothermal magnetization data $M(B) \parallel [100]$ at 1.8 K as displayed in Fig. 2(b). The curvatures of $M(B)$ are gradually suppressed as decreasing $x$. To evaluate the intersite magnetic interaction, we use a mean-field Hamiltonian expressed as

$$\mathcal{H} = \mathcal{H}_{\mathrm{CEF}} + g_J \mu_{\mathrm{B}} \boldsymbol{J}\boldsymbol{B} - K\langle \boldsymbol{J}\rangle \boldsymbol{J}, \tag{2}$$

where $g_J = 8/11$ is the Landé $g$-factor for a Nd$^{3+}$ ion, $J$ the total angular momentum, and $K$ the strength of the intersite magnetic interaction between the Nd moments. The solid line represents the CEF calculation with no magnetic interaction as described below.

$$M = \frac{g_J \mu_{\mathrm{B}}}{Z} \sum_i \langle i|J|j\rangle e^{-E_i/k_{\mathrm{B}}T}. \tag{3}$$

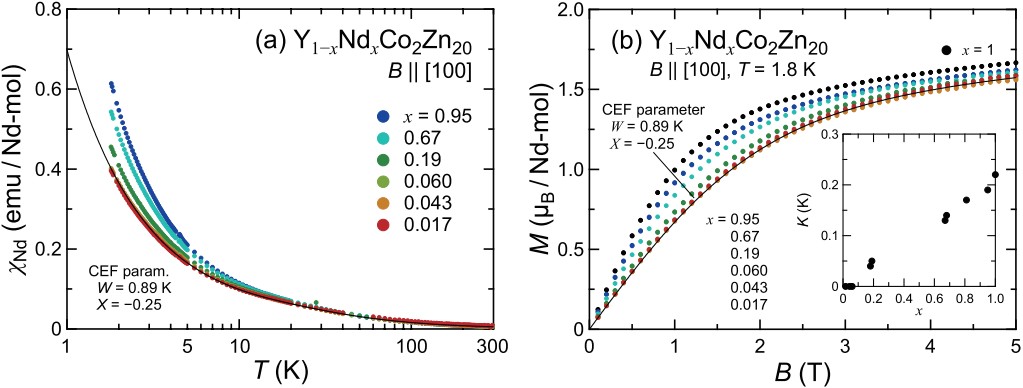

Figure 2: (a) $4f$ contribution of the magnetic susceptibility $\chi_{\mathrm{Nd}}$ versus temperature $T$ of Y$_{1-x}$Nd$_x$Co$_2$Zn$_{20}$ for $0.017 \leq x \leq 0.95$ on a logarithmic scale. The solid line shows the $\chi(T)_{\mathrm{Nd}}(T)$ data calculated with the CEF parameters $W = 0.89$ K and $X = -0.25$. (b) Isothermal magnetization $M(B)$ at $T = 1.8$ K, where the data for $x = 1$ are taken from [8]. The data are normalized by the Nd composition $x$. The solid line was calculated using the CEF parameters [17]. The inset shows the intersite magnetic interaction parameter $K$ estimated to reproduce the $M(B)$ data.

$Z$ is the partition function. The calculation with $K = 0$ reproduces the data for $x \leq 0.06$. The intersite magnetic interaction $K$ versus $x$ is plotted in the inset of Fig. 2(b). The values of $K$ for $x \leq 0.06$ were estimated to be zero within the error of the present analysis. Ferro-type intersite magnetic interaction deduced from the $M(B)$ data is consistent with the positive value of $\theta_{\mathrm{p}}$ evaluated from the $\chi(T)$ data below 10 K. These facts suggest that the single-site state of the $\Gamma_6$ doublet with no intersite magnetic interaction is realized in $\mathrm{Y}_{1-x}\mathrm{Nd}_x\mathrm{Co}_2\mathrm{Zn}_{20}$ for $x \leq 0.06$. Therefore, this dilute Nd system could provide a good platform to investigate the single-site hybridization effect of the $4f$ electrons with the conduction electrons. The low-temperature transport and magnetic properties below 1.8 K will be reported in a forthcoming paper.

## 4 Conclusion

We have measured $\rho(T)$, $C(T)$, $\chi(T)$, and $M(B)$ of $\mathrm{Y}_{1-x}\mathrm{Nd}_x\mathrm{Co}_2\mathrm{Zn}_{20}$ for $0.017 \leq x \leq 0.95$. The Schottky anomalies of $C(T)$ at around 13 K are moderately reproduced with the CEF level scheme of $\mathrm{NdCo}_2\mathrm{Zn}_{20}$. The intersite magnetic interaction parameter $K$ is estimated from the $\chi(T)$ and $M(B)$ data. The positive values of $K$ for $x \geq 0.18$ decrease to almost zero for $x \leq 0.06$. Thus, the diluted Nd system $\mathrm{Y}_{1-x}\mathrm{Nd}_x\mathrm{Co}_2\mathrm{Zn}_{20}$ for $x \leq 0.06$ serves as a model system to study the on-site interaction of $\Gamma_6$ doublet ground state of the $4f$ electrons with conduction electrons.

## Acknowledgements

The authors thank Y. Shibata for the electron-probe microanalysis carried out at N-BARD, Hiroshima University. The measurements with MPMS and PPMS were performed at N-BARD, Hiroshima University.

**Funding information** This work was financially supported by grants-in-aid from MEXT/JSPS of Japan [Grants No. JP26707017, No. JP15H05886 (J-Physics), No. JP18H01182, and No. JP21J12792].

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
