# Peer review of "Transport and Magnetic Properties in the Nd Diluted System Y$_{1-x}$Nd$_{x}$Co$_{2}$Zn$_{20}$"

_SciPost Physics Proceedings, doi:SciPost Phys. Proc. 11, 010 (2023)_

## Round 2 · Author Response

We thank the referee for appreciating our study of transport and magnetic properties in the Nd diluted system Y1−xNdxCo2Zn20. Our responses to all the comments are itemized in the list of changes. Changes in the manuscript are shown in red. We believe that the manuscript could be improved for the publication.

---

## Round 2 · List of Changes

1. Some comments on crystal structural data of the compounds under study could be made in the beginning of the results and discussion. With Nd-substitution at Y-site, how the lattice parameters vary.

We determined the Nd composition of each sample from the magnetization data at T = 1.8 K comparing the calculated value with the CEF parameters W = 0.89 K and X = −0.25.
The Nd composition dependence of the lattice parameter linearly increase as increasing x, and this result ensures the homogeneity of the Nd composition.
We have added a sentence as follows.

p. 2, line 47-48
The lattice parameter determined by the Rietveld analysis increases linearly with respect to x.

2. A remark on band filling effects in Co could be useful while introducing YCo2Zn20 compound as a non-magnetic analogue.

Magnetic susceptibility of YCo2Zn20 is reported to be temperature independent, yielding Pauli paramagnetism [19]. The electronic structure calculation of YCo2Zn20 suggests an intermetallic state with negligible electron-electron correlation. Thereby, the ground state becomes non-magnetic.

We have added a sentence as follows.

p. 3, line 84-85
It is noted that the band structure calculation of YCo2Zn20 suggests an intermetallic state with negligible electron-electron correlation, leading to a non-magnetic ground state [19,20].

[19] S. Jia et al., Phys. Rev. B 77, 104408 (2008).
[20] M. Cabrera-Baez et al., Phys. Rev. B 92, 214414 (2015).

3. In figure 1b inset, better to explicitly mark the X-axis label and the corresponding text in page 3 para 2 could read as '....Cm/T vs T...'.

Thank you for your suggestion. We have modified the Fig. 1b and revised the sentence.

p. 3, line 70-71
The magnetic specific heat divided by temperature as a function of temperature, Cm(T) / T vs T, is shown in the inset of Fig. 1(b).

4. Inelastic neutron scattering data are not published yet. Therefore a comparison made in page 3 discussion is fine. However, it need not be emphasized in conclusion paragraph.

We really appreciate your comment. We have revised the conclusion paragraph as follows.

p. 5, line 110-112
We have measured ρ(T), C(T), χ(T), and M(B) of Y1−xNdxCo2Zn20 for 0.017 ≤ x ≤ 0.95. The Schottky anomalies of C(T) at around 13 K are moderately reproduced with the CEF level scheme determined for NdCo2Zn20.

---

## Editorial Decision

published